# Production of deoxycholic acid by low-abundant microbial species is associated with impaired glucose metabolism

Annika Wahlström[1], Ariel Brumbaugh[2,3,4,5], Wilhelm Sjöland[1], Lisa Olsson [1], Hao Wu[6], Marcus Henricsson [1], Annika Lundqvist[1], Kassem Makki[1], Stanley L. Hazen [7,8,9], Göran Bergström [1,10], Hanns-Ulrich Marschall [1,12], Michael A. Fischbach [2,3,4,5] ✉ & Fredrik Bäckhed [1,10,11] ✉

Alterations in gut microbiota composition are suggested to contribute to cardiometabolic diseases, in part by producing bioactive molecules. Some of the metabolites are produced by very low abundant bacterial taxa, which largely have been neglected due to limits of detection. However, the concentration of microbially produced metabolites from these taxa can still reach high levels and have substantial impact on host physiology. To explore this concept, we focused on the generation of secondary bile acids by 7α-dehydroxylating bacteria and demonstrated that addition of a very low abundant bacteria to a community can change the metabolic output dramatically. We show that *Clostridium scindens* converts cholic acid into the secondary bile acid deoxycholic acid (DCA) very efficiently even though the abundance of *C. scindens* is low, but still detectable by digital droplet PCR. We also show that colonization of germ-free female mice with a community containing *C. scindens* induces DCA production and affects host metabolism. Finally, we show that DCA correlates with impaired glucose metabolism and a worsened lipid profile in individuals with type 2 diabetes, which implies that this metabolic pathway may contribute to the development of cardiometabolic disease.

Cardiometabolic diseases such as obesity and type 2 diabetes mellitus (T2D) are increasing globally and there is an urge for new treatments. Alterations in the gut microbiota are implicated in metabolic diseases[1], but how these alterations contribute to the disease is still not fully understood. One key challenge in microbiota research is to understand how different bacteria function in consortia to produce bioactive molecules. Some microbial metabolites are produced by very low abundant bacterial taxa, yet the metabolites reach very high

[1]Wallenberg Laboratory and Department of Molecular and Clinical Medicine, Institute of Medicine, Sahlgrenska Academy, University of Gothenburg, Gothenburg, Sweden. [2]Department of Bioengineering, Stanford University, Stanford, CA, USA. [3]Department of Microbiology and Immunology, Stanford University School of Medicine, Stanford University, Stanford, CA, USA. [4]ChEM-H Institute, Stanford University, Stanford, CA, USA. [5]Chan Zuckerberg Biohub, San Francisco, CA, USA. [6]State Key Laboratory of Genetic Engineering, Human Phenome Institute, Fudan Microbiome Center, and Department of Bariatric and Metabolic Surgery, Huashan Hospital, Fudan University, Shanghai, China. [7]Department of Cardiovascular & Metabolic Sciences, Lerner Research Institute, Cleveland, OH, USA. [8]Center for Microbiome and Human Health, Cleveland Clinic, Cleveland, OH, USA. [9]Department of Cardiovascular Medicine, Heart, Vascular and Thoracic Institute, Cleveland Clinic, Cleveland, OH, USA. [10]Region Västra Götaland, Sahlgrenska University Hospital, Department of Clinical Physiology, Gothenburg, Sweden. [11]Novo Nordisk Foundation Center for Basic Metabolic Research, Faculty of Health Sciences, University of Copenhagen, Copenhagen, Denmark. [12]Deceased: Hanns-Ulrich Marschall. ✉e-mail: fischbach@fischbachgroup.org; Fredrik.Backhed@wlab.gu.se

concentrations in the gut and circulation where they can have substantial impact on host physiology[2]. An example of such metabolites are secondary bile acids, which are among the most abundant microbiota-derived molecules, although only a few known bacterial species are able to produce them. Bile acids signal via host nuclear- and G-protein coupled receptors and can affect many different metabolic pathways[3]. Different bile acids have varying affinities for their receptors; the gut microbiota can thus modulate signaling via the host receptors by changing the bile acid composition[4–6]. The gut microbiota modifies the bile acid pool by metabolizing primary bile acids into secondary bile acids through dehydroxylation, dehydrogenation and epimerization[7]. The two most prevalent secondary bile acids in humans are deoxycholic acid (DCA) and lithocholic acid (LCA), which are produced by 7α-dehydroxylating bacteria from cholic acid (CA) and chenodeoxycholic acid (CDCA), respectively[5]. Microbial 7α-dehydroxylation is a complex multistep process, which has just recently been fully unveiled and shown to require enzymes encoded by the *bai*-operon[7,8]. The number of bacterial species that possess the whole *bai*-operon are limited and they have been identified in *Eubacterium* and *Clostridium* cluster XIVa and XI[7–11]. Although, 7α-dehydroxylating bacteria are typically present at low abundance in the human gut, their metabolic potential is high and more than 95% of the bile acid pool in the large intestine consists of dehydroxylated bile acids[7,12]. Furthermore, in addition to DCA and LCA there is a wide range of other microbially produced bile acid metabolites such as iso-, epi-, and oxo-bile acids, which might be biologically important but are less well studied[12–14].

In this study, we explore if addition of a low abundance bacterial species into a community could have significant impact on the metabolic output. We chose to focus on bile acids since they are involved in many physiological processes and secondary bile acids are among the most abundant microbial-derived metabolites. We investigated DCA production by a low abundant bacterial species, *Clostridium scindens*, both in vitro and in vivo, and demonstrated a link between DCA and impaired host metabolism.

## Results

### Generation of a simplified community with bile acid-metabolizing capacity

To investigate if we could produce a system to assess the function of low abundant bacteria, we first established an in vitro system in which a 7α-dehydroxylating bacterial species was present at low abundance, to study the formation of DCA from CA. First, we constructed a 'base' synthetic community, based on previous work[15] consisting of bacterial species that lack the *bai* genes. The nine selected bacteria were cultured individually in the presence of CA to assess their bile acid-metabolizing capacity. Three of them (*Bacteroides caccae*, *Bacteroides thetaiotaomicron* and *Bacteroides ovatus*) produced 7-oxoCA, while none of the base community members produced DCA (Fig. 1A). Similarly, the nine bacteria did not produce DCA in vitro when they were cultured together (Fig. 1B). In contrast, *C. scindens*, a known 7α-dehydroxylating bacterium[7,16–18], converted CA to DCA both when cultured alone (Fig. 1C) and together with the base community (Fig. 1D).

Finally, to mimic the situation in the human gut we performed serial dilutions to decrease the abundance of *C. scindens* in the simplified community. To quantify the level of *C. scindens* in our diluted communities we used shotgun metagenomic sequencing coupled with digital droplet PCR, with specific primer-probe sets for *C. scindens* and *B. caccae*, a high abundance community member detected in the shotgun metagenomic data and used as a reference (Supplementary Table 1). We then assessed the capability of *C. scindens* to generate DCA in the different dilutions and we found that at dilutions that range from 1:10⁴ to 1:10⁶, *C. scindens* could still convert 100 µM of CA to DCA

in 24 h, establishing that in a controlled synthetic community setting, a low abundant organism can have a very high metabolic output (Fig. 1E).

### DCA-producing *C. scindens* modulates physiology and glucose metabolism in colonized mice

We next investigated if the base community +/− *C. scindens* produced DCA in vivo by colonizing germ-free mice for two weeks. Since germ-free mice have low levels of CA[4], the precursor of DCA, we fed the mice a diet containing 1% CA. At termination, we analyzed microbiota composition in the inoculum and in caecum by performing shotgun metagenomic sequencing (Fig. 2A). We observed that the caecal microbiota was dominated by *Bacteroides* species and that the relative abundance of *C. scindens* was below the limit of quantification. Hence, we performed digital droplet PCR and found that the ratio of *C. scindens* genome copies to the total community was 1:14,397 (Fig. 2B). Importantly, *C. scindens* was not detected in any of the mice colonized with the base community, establishing that we could generate communities that differed in one low abundant species in vivo.

We then analyzed bile acids in serum from vena cava and caecal content from mice colonized with the base community and demonstrated that these mice could deconjugate tauro-beta-muricholic acid (TβMCA) and tauro-CA (TCA) into βMCA and CA respectively, but they did not further metabolize them to secondary bile acids (Fig. 2C, D). In contrast, mice colonized with base community + *C. scindens* showed substantial DCA production despite low abundance of *C. scindens* (0.04 ± 0.01% of all community members). DCA levels ranged from 3186–9183 nM in serum and 499–1091 pmol/mg in caecum of mice colonized with base community + *C. scindens*, while DCA was undetectable in serum of mice colonized with base community and only traces were found in caecum (Fig. 2C, D). To compare the DCA levels in our colonized mice with physiological levels we also measured DCA in serum and caecum of conventionally raised mice on chow or 1% CA-diet. The conventionally raised mice on CA-diet had equally high DCA levels in serum as mice colonized with base community + *C. scindens* but higher DCA levels in caecum (Figure S1A). In concordance with previous findings, the CA-diet resulted in elevated serum levels of alanine transferase (ALT) and aspartate aminotransferase (AST), indicating a hepatotoxic effect of CA[19]. However, there were no differences between mice colonized with base community and base community + *C. scindens* (Figure S1B). Of note, mice colonized with *C. scindens* had reduced hepatic gene expression of *Cyp7a1* while there was no significant difference in *Cyp8b1* expression between the two groups (Figure S1C).

Furthermore, mice colonized with base community + *C. scindens* had similar body weight as mice colonized with base community (Fig. 2E). However, mice colonized with base community + *C. scindens* had increased adiposity (Fig. 2F) and liver weight (Fig. 2G), but reduced caecum weight (Fig. 2H), compared to mice colonized with base community. In line with increased adiposity, liver triglycerides were also increased in mice colonized with base community + *C. scindens* and correlated with relative liver weight (Figure S1D). Analysis of caecal short chain fatty acids showed higher butyrate levels in mice colonized with base community + *C. scindens*, which could be due to increased carbohydrate fermentation and may in part explain the reduction in relative caecum weight (Figure S1E).

At the end of the experiment, we performed an intraperitoneal insulin tolerance test and observed that mice colonized with base community + *C. scindens* had impaired insulin tolerance, indicating that glucose metabolism may be altered (Fig. 2I) although there were no differences in fasting blood glucose or fasting insulin (Fig. 2J, K). Taken together, our in vitro and in vivo systems show that addition of a low abundance bacterial species into a community can have a large effect on the microbial metabolic output, which subsequently can alter host metabolism.

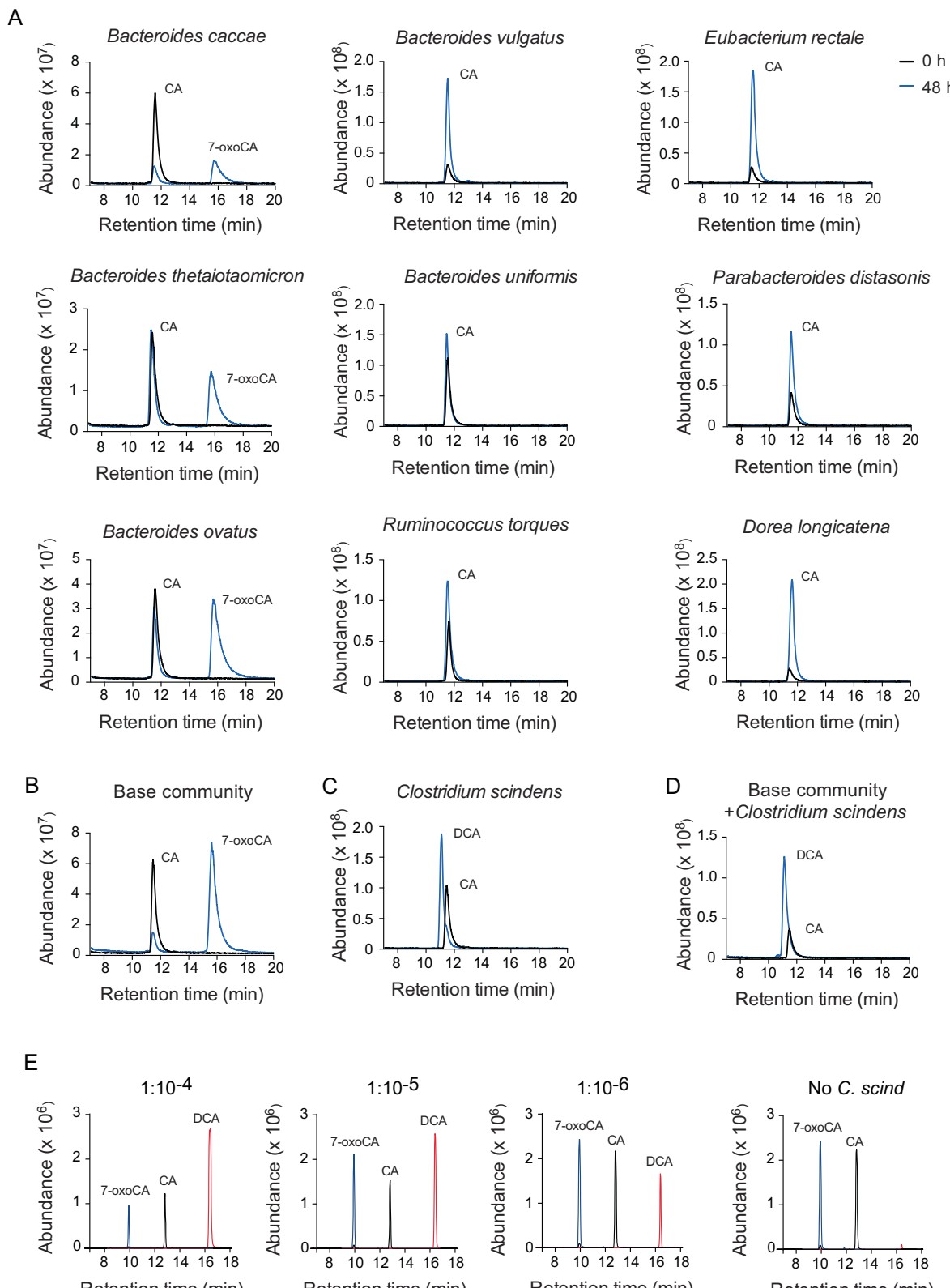

## DCA is enriched in T2D individuals and correlates with worsened metabolic characteristics

To investigate if alterations in DCA levels could be implicated in human metabolic diseases, we selected 100 individuals with normal glucose tolerance (NGT) and 100 individuals with treatment naive T2D from the IGT Microbiota and SCAPIS cohorts and analyzed their bile acid profiles. The NGT and T2D groups were matched for sex, age, weight,

BMI, statins, and proton-pump inhibitor use and none in the T2D group were on diabetes medication (Table 1).

Bile acid analysis in plasma revealed that total bile acid levels were increased in individuals with T2D compared with NGT controls (2083 nM ± 170 in the T2D group vs 1231 nM ± 91 in the NGT group, $p < 0.0001$; Fig. 3A and Supplementary Data 1). Analysis of individual bile acids demonstrated that 12-hydroxylated secondary bile acids e.g.,

**Fig. 1 | In vitro assessment of DCA formation by the bacterial species included in the simplified communities. A–D** Assessment of DCA formation after 48 h culture in Mega Medium containing 100 µM CA by the base community members *Bacteroides caccae* ATCC 43185, *Bacteroides vulgatus* ATCC 8482, *Eubacterium rectale* ATCC 33656, *Bacteroides thetaiotaomicron* VPI-5482, *Bacteroides uniformis* ATCC 8492, *Parabacteroides distasonis* ATCC 8503, *Bacteroides ovatus* ATCC 8483, *Ruminococcus torques* ATCC 27756 and *Dorea longicatena* DSM 13814 cultured individually (**A**), base community members cultured together (**B**), *Clostridium* *scindens* ATCC 35704 cultured individually (**C**) and base community + *Clostridium scindens* cultured together (**D**). **E** Assessment of DCA production after 24 h in serial dilutions of *Clostridium scindens* in the base community spiked with CA from a 100 mM stock solution to a final concentration of 100 µM. Abundance of CA, DCA and 7-oxoCA was measured by GC-MS in (**A–D**) and by UPLC-Q-TOF-MS in (**E**). CA, cholic acid; DCA, deoxycholic acid; 7-oxoCA, 7-oxocholic acid. Black line indicates bile acid abundance at 0 h and blue line indicates bile acid abundance at 48 h (**A–C**); blue line indicates 7-oxoCA; black line indicates CA and red line indicates DCA (**E**).

DCA, isoDCA and 12-epiDCA were the most significantly increased bile acids in individuals with T2D (Fig. 3B–D and Supplementary Fig. 2A) and 8 additional bile acids were also increased (Supplementary Fig. 2A and Supplementary Data 1).

Next, we analyzed bile acid composition in faeces but did not observe any significant differences in total bile acid levels (Fig. 3E) or in any of the analyzed bile acid species (Supplementary Fig. 2B and Supplementary Data 2). However, DCA was the most abundant bile acid and tended to be increased in faeces from individuals with T2D ($p = 0.06$ before adjustment for multiple testing) (Fig. 3F and Supplementary Data 2) and correlated with DCA levels in plasma (Fig. 3G).

To investigate associations between DCA and physiological parameters related to impaired glucose metabolism, we correlated clinical parameters (Table 1) with bile acids in faeces and plasma. Although we did not observe significant differences in faecal DCA between the NGT and T2D groups, we observed significant correlations between faecal DCA and parameters associated with impaired glucose metabolism such as HOMA-IR, fasting blood glucose (FBG), HbA1c, and insulin (Fig. 3H and Supplementary Data 2). In addition, DCA correlated positively with triglycerides (TG) and negatively with high-density lipoprotein (HDL), which is in line with a worsened lipid profile. The correlations were observed when all individuals were analyzed together and also when individuals with T2D were analyzed separately (Fig. 3H). In contrast, we did not observe any significant correlations when the NGT group was analyzed separately. DCA also correlated positively with C-reactive protein (CRP) and white blood cells (e.g., neutrophils) suggesting that it is associated with inflammation. Plasma DCA showed a similar correlation pattern as faecal DCA and the strongest correlation was found with HOMA-IR, both in the total cohort and when the T2D group was analyzed separately (Fig. 3H and Supplementary Data 1). In summary, DCA, in both faeces and plasma, is associated with impaired glucose metabolism and a worsened lipid profile.

## Discussion

In this study we demonstrated that a low abundance bacterial species, which might be neglected due to detection limitations, can have a significant impact on the production of bioactive metabolites.

We showed that *C. scindens*, although present at low abundance, could produce large amounts of DCA and affect insulin tolerance in colonized mice, suggesting that DCA may contribute to impaired glucose metabolism. Furthermore, we demonstrated that DCA, in plasma and faeces, correlates with impaired glucose metabolism and a worsened lipid profile in individuals with T2D. Our findings highlight that low abundant bacterial species may have profound effects on metabolite levels in the gut and circulation and should receive more attention. One challenge of studying these organisms in metagenomes is that they may be close to the detection limit and thus robust analyses are lacking. Accordingly, this concept may be extended to other metabolites that are produced by low abundant taxa and emphasize the importance of studying gut microbial metabolites in the circulation of host organisms. Another challenge is that bile acid transformation is very complex and a variety of environmental factors, such as medications, diet, pH, gas atmosphere and presence of enzymatic cofactors can influence interactions between bile acids and bacteria in the gut[20]. These parameters are dependent on the crosstalk between different bacteria in a community and the function of a specific bacterial species can differ when it is cultured alone compared to when it is cultured in a consortium of other bacteria. There can also be discrepancies in bile acid transformation by bacteria in vitro and in vivo[21,22], which emphasize the importance of comprehensive studies on bile acid metabolizing bacteria.

Our results clearly show that very low abundance of 7α-dehydroxylating bacteria is sufficient to produce large amounts of DCA and this supports the idea that 7α-dehydroxylation could be exclusive for very few bacteria in the human gut. However, DCA levels in caecum were lower than the levels in conventionally raised mice, which have a complete microbiota, and we cannot exclude that there are other unidentified bacterial species in the human gut that contribute to the formation of DCA, such as *Clostridium hylemonae*[23], and further studies are needed to provide an extended view of bile acid metabolizing bacteria. There could also be synergistic effects between *C. scindens* and other bacteria in the gut, comparable to what we previously have shown between *Faecalibacterium prausnitzii* and *Desulfovibrio piger*[24]. In that study, co-colonization of mice with *D. piger* increased the abundance and metabolic capacity of *F. prausnitzii* and it will be important to elucidate similar mechanisms for *C. scindens* and other DCA-producing bacteria.

Bile acids have generated significant interest as signaling molecules over the past years and here we demonstrate that insulin tolerance is affected in mice colonized with DCA-producing *C. scindens*. This is in line with a previous study where Zaborska et al showed that DCA supplementation in mice leads to impaired glucose homeostasis (both by insulin- and glucose tolerance tests) and increased hepatic ER stress and decreased hepatic insulin signaling[25]. In another study, Yoshimoto and co-workers showed that DCA production was increased in mice with diet- or genetically induced obesity and that DCA contributed to cancer development in the obese mice[26]. Contrary to Yoshimoto's study, we used a chow diet supplemented with CA and our mice were not obese. Nevertheless, we found increased adiposity and liver triglycerides in our DCA-producing mice, supporting that there is a link between DCA and lipid accumulation although the mechanisms might be different.

Bile acid synthesis has also been implicated in metabolic changes and we observed decreased expression of *Cyp7a1* in the livers of mice colonized with base community + *C. scindens* but no difference in the expression of *Cyp8b1*. However, the contribution of endogenously synthesized bile acids in our mouse model is probably negligible since the mice were administered a very high dose of exogenous CA through the diet.

We also showed that DCA levels are increased in plasma of individuals with T2D and that the altered bile acid profile is not a consequence of diabetes medication since all individuals with T2D were newly diagnosed and had not been treated for their diabetes. It is currently unclear why the faecal concentrations of DCA were not significantly different, but one possible explanation could be that we only obtained spot samples rather than 24-h samples, or that there are differences in bile acid uptake.

Previous studies have shown conflicting results regarding plasma bile acid profiles and glucose metabolism[27]. Hausler et al showed that an increased ratio of 12αOH/non12αOH bile acids is associated with insulin resistance[28], while another study found no correlation between

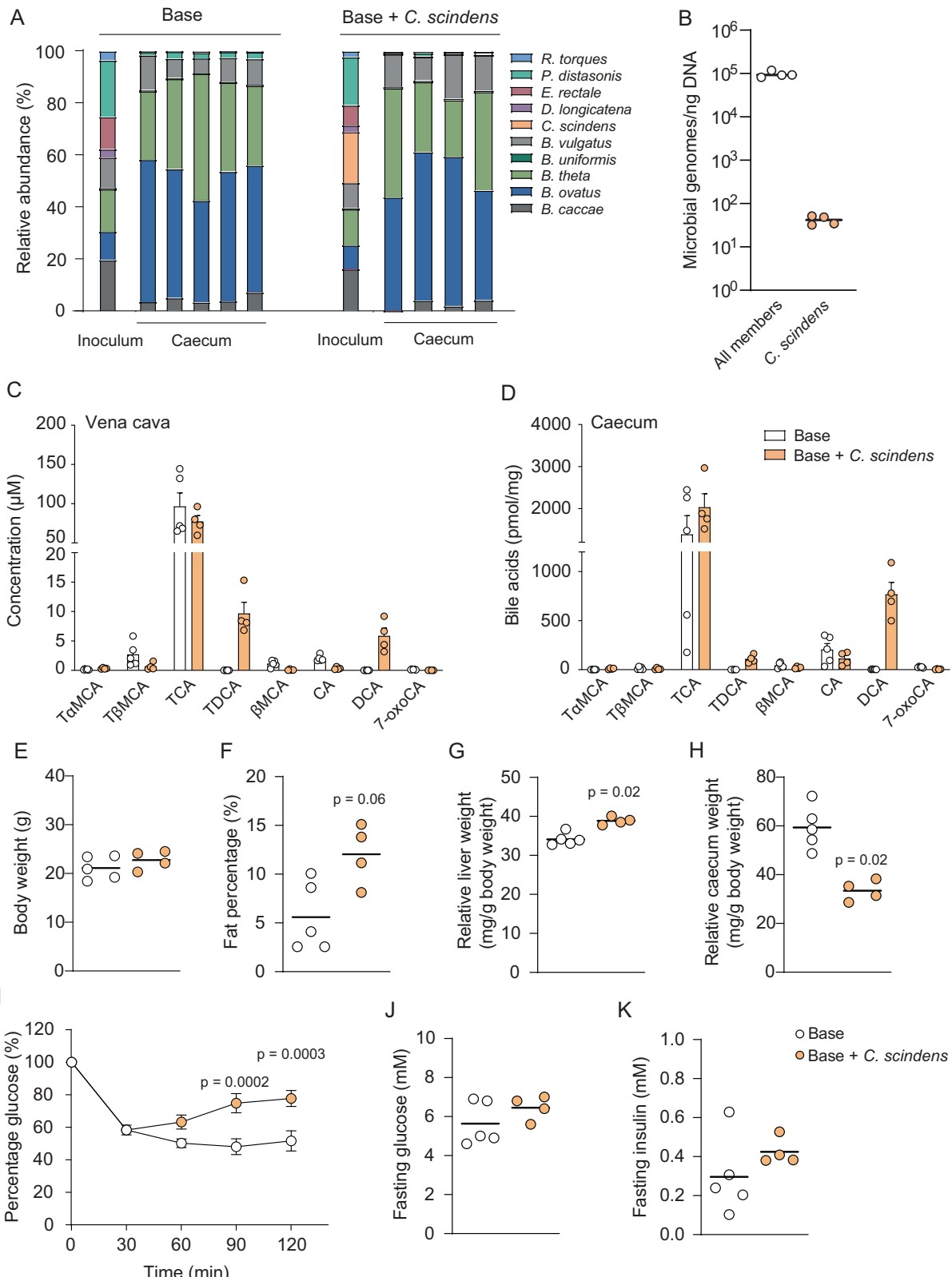

12αOH/non12αOH ratio and HOMA-IR or risk of new onset of diabetes[27]. One potential reason for this discrepancy might be that there are differences in the microbial conversion within the 12αOH bile acid species i.e., conversion of CA into DCA and/or iso-DCA and 12-epiDCA, hence 12αOH/non12αOH ratio might be a crude measurement and needs to be further dissected. We found strong correlations

between DCA and parameters related to glucose and lipid metabolism and our clinical findings suggest that DCA may be a biomarker for inflammation and cardiometabolic diseases. However, further studies are needed to establish the causality between DCA and metabolic disease and prospective studies may reveal if DCA can be used to identify individuals at risk of developing T2D.

**Fig. 2 | Colonization of germ-free mice with bacterial communities. A** Relative abundance of bacteria in the inoculum and the caecal contents of mice colonized with base community or base community + *C. scindens* assessed by shotgun metagenomic sequencing. **B** Microbial genome copies per nanogram bacterial DNA in the caecal content of mice colonized with base community + *C. scindens* assessed by digital droplet PCR. The base community members were quantified using primer-probe sets for the gene *tilS*, and *C. scindens* was quantified using primer-probe sets for the *baiCD* gene (see Supplementary Table 1). **C, D** Bile acid levels in vena cava (**C**) and caecum (**D**) of mice colonized with base community or base community + *C. scindens* measured by UPLC-MS/MS. **E–H** Physiological parameters of the mice including body weight (**E**), adiposity (**F**), relative liver weight (**G**) and relative caecum weight (**H**) measured 2 weeks after colonization. **I** Insulin tolerance test performed by intraperitoneal injection of insulin. Glucose was measured at 0, 30, 60, 90, 120 min after injection and data is presented as percentage of glucose compared to the 0 min time point. **J, K** Fasting blood glucose (**J**) and insulin (**K**) were measured at the 0 min time point. *n* = 5 samples for the base community group (indicated by white color in **B–K**) and *n* = 4 samples for the base community + *C. scindens* group (indicated by orange color in **B–K**); *P*-values indicate differences between mice colonized with base community and base community + *C. scindens* analyzed with two-tailed Mann–Whitney (**E–H** and **J, K**) or 2-way ANOVA with Šidák multiple comparisons test (**I**). Error bars indicate standard error of the mean (**C, D** and **I**). CA cholic acid, DCA deoxycholic acid, βMCA beta-muricholic acid, TCA tauro-cholic acid, TDCA tauro-deoxycholic acid, TαMCA tauro-alpha-muricholic acid, TβMCA tauro-beta-muricholic acid, 7-oxoCA 7-oxocholic acid.

## Methods

### Bacterial strains and culture conditions

Synthetic community strains were cultured in pre-reduced Mega Medium[15] or on EG Agar[29] at 37 °C in an anaerobic chamber (Coy Laboratories, Grass Lake, MI, USA) with an atmosphere of 5% hydrogen, 10% CO$_2$ and 85% N$_2$. Synthetic community strains included *Bacteroides uniformis* ATCC 8492, *Bacteroides vulgatus* ATCC 8482, *Bacteroides thetaiotaomicron* VPI-5482, *Bacteroides caccae* ATCC 43185, *Bacteroides ovatus* ATCC 8483, *Parabacteroides distasonis* ATCC 8503, *Eubacterium rectale* ATCC 33656, *Dorea longicatena* DSM 13814, *Ruminococcus torques* ATCC 27756, and *Clostridium scindens* ATCC 35704. All strains were obtained from the American Type Culture Collection (ATCC), except *D. longicatena* DSM 13814, which was obtained from The Leibniz Institute DSMZ (Braunschweig, Germany).

### Animal experiments

Germ-free female C57Bl/6 mice were maintained in flexible plastic gnotobiotic isolators under a strict 12 h light cycle (light from 7 a.m. to 7 p.m.), 20 ± 1 °C temperature, air humidity of 45–70%, under standard germ-free conditions, and fed an autoclaved chow diet (Lab diet, St Louis, MO; #5021) supplemented with 1% CA (Envigo Teklad Diets, Madison, WI) ad libitum. Germ-free isolators were routinely tested for sterility by culturing and PCR analysis of faeces amplifying the 16S rRNA gene.

Germ-free mice were colonized after 4 h fasting by gavage with 0.2 ml of the cultured communities (base community or base community including *C. scindens*). Before colonization all the mice were weighed and randomized into two weight-matched groups. The communities of bacteria were thawed and gavaged into the germ-free mice within 2 h after thawing. The colonization was repeated after 2 days. Mice were colonized at 17 weeks of age and maintained for 2 weeks after colonization. Gut microbial composition following colonization was monitored by sequencing and digital droplet PCR (ddPCR) as detailed below.

An insulin tolerance test was performed by injecting insulin 0.5 U/kg body weight intraperitoneally after 4 h fasting. Tail blood samples were collected at 0, 30, 60, 90, and 120 min and blood glucose levels were determined using a glucose meter (Contour NEXT, Ascensia Diabetes Care, Stockholm, Sweden). Body composition was determined with magnetic resonance imaging (EchoMRI, Houston, TX) after the insulin tolerance test and blood was collected from the inferior vena cava under deep isoflurane-induced anesthesia. The mice were then killed, and serum and tissues were collected. Body, liver and caecum weights were recorded, and tissues were immediately frozen in liquid nitrogen and stored at -80 °C until further analyzed.

In a separate experiment conventionally raised female C57Bl/6 mice were maintained under a strict 12 h light cycle (light from 7 a.m. to 7 p.m.), 20 ± 1 °C temperature, air humidity of 45–70%, under standard specific-pathogen-free conditions and fed either chow diet (Lab diet, St Louis, MO) or a diet supplemented with 1% CA (Envigo Teklad Diets, Madison, WI) ad libitum for two weeks. The mice were then killed, and serum and caecum were collected for analyses. All animal experiments were performed at Experimental Biomedicine, University of Gothenburg, using protocols approved by the Gothenburg Animal Research Ethics Committee (Ethical number 4805/23).

### Human cohorts

200 individuals (100 with treatment naive T2D and 100 BMI matched controls with normal glucose tolerance (NGT)) were selected from the IGT Microbiota cohort (45 T2D and 45 NGT), and the Swedish Cardiopulmonary Bioimage Study (SCAPIS)-Gothenburg cohort (55 T2D and 55 NGT). The design of the two human cohorts have been described in detail elsewhere[30,31]. Both cohorts comprise men and women aged 50–64 years from the Gothenburg area, Sweden, who were recruited at random from the census register. Exclusion criteria were known diabetes; inflammatory diseases, such as Crohn's disease, ulcerative colitis, rheumatic diseases; treatment with steroids or immunomodulatory drugs; cancer (unless relapse free for the preceding 5 years); cognitive dysfunction; and treatment for infectious diseases and with antibiotics in the past three months. Accordingly, none of the age, sex and BMI matched controls were taking either metformin or weight reduction medications like GLP1 agonists. Individuals were also excluded if they did not understand Swedish and if they were born outside Sweden. All participants gave informed consent, and the study was approved by the Ethics Review Board in Gothenburg (560-13) and in Umeå 2010-228-31M (2012-285-32M and 2014-33-32).

We collected plasma and faecal samples from the 200 individuals and metabolic parameters were evaluated. Bile acids were assessed in plasma and faeces.

### DNA extraction and sequencing of bacterial cultures and mouse caecal samples

Total DNA from mouse caecal contents and bacterial culture pellets were extracted using the PowerFecal DNA Isolation kit (MoBio, Carlsbad, CA, USA), and quantified using the Qubit dsDNA BR Assay Kit and Invitrogen Qubit Fluorometer (Thermo Fischer, Waltham, MA, USA). Sequencing libraries were generated using the Nextera XT DNA Library Preparation kit and Index v2 kit (Illumina, Hayward, CA, USA) and sequenced on an Illumina MiSeq with MiSeq Reagent Kit v2 (300 cycles) (Illumina, Hayward, CA, USA). Sequencing reads were mapped to a reference database built with only the genomes of bacterial strains in the study using Bowtie2 2.2.9[32] and the relative abundance of bacterial community members was quantified using SAMtools 1.3.1[33].

### ddPCR quantification

IDT Taqman ddPCR primer/probe sets were designed to the *baiCD* gene (bile acid-inducible gene) for *C. scindens* ATCC 35704, and the single copy essential gene *tilS* (tRNAIle-lysidine synthase) for *B. caccae* ATCC 43185 to ensure each genome was counted only once. In brief, a 25 μL reaction comprising 9 μL water, 12.5 μL 2× BioRad ddPCR Supermix for Probes (no dUTP) [cat: 186-3023], 1.25 μL NEB HindIII-HF, 1.25 μL Taqman Primer/Probe Set 20X in IDTE buffer, 1.25 μL input DNA was used for each sample. Droplets were generated with the QX200

**Table 1 | Characteristics of the study participants in the NGT and T2D groups**

| Characteristics of study participants | | | |
|---|---|---|---|
| | **NGT** | **T2D** | ***P*-value** |
| Gender | 44 (% females) | 42 (% females) | ns |
| Age (years) | 57.8 ± 4.5 (*n* = 100) | 58.7 ± 4.4 (*n* = 100) | ns |
| Weight (kg) | 94.3 ± 17.6 (*n* = 100) | 94.6 ± 18.7 (*n* = 100) | ns |
| Height (cm) | 174.2 ± 10 (*n* = 100) | 173.6 ± 8.4 (*n* = 100) | ns |
| BMI (kg/m²) | 30.9 ± 4.1 (*n* = 100) | 31.2 ± 5.1 (*n* = 100) | ns |
| Waist (cm) | 104 ± 12.7 (*n* = 100) | 108.9 ± 13.6 (*n* = 100) | 0.01 |
| Hip (cm) | 111 ± 9.6 (*n* = 100) | 109.2 ± 10.4 (*n* = 100) | ns |
| whr | 0.9 ± 0.1 (*n* = 100) | 1 ± 0.1 (*n* = 100) | 5.2e−6 |
| SBP (mmHg) | 131 ± 17.9 (*n* = 100) | 133.2 ± 17.0 (*n* = 99) | ns |
| DBP (mmHg) | 81 ± 10.7 (*n* = 100) | 81.8 ± 10.4 (*n* = 99) | ns |
| Hb (g/L) | 142.8 ± 10.9 (*n* = 99) | 146.1 ± 12.0 (*n* = 98) | 0.05 |
| FBG (mmol/L) | 5.5 ± 0.5 (*n* = 96) | 8.3 ± 3.0 (*n* = 100) | 1.5e−15 |
| OGTT 2 h BG (mmol/L) | 6.4 ± 1.1 (*n* = 100) | 13.8 ± 1.5 (*n* = 26) | 2.6e−22 |
| Insulin (pmol/L) | 26.3 ± 26.0 (*n* = 100) | 38.1 ± 41.1 (*n* = 100) | 0.02 |
| HbA1c (%) | 34.9 ± 3.1 (*n* = 100) | 48.8 ± 17.4 (*n* = 98) | 4.6e−12 |
| HOMA_IR | 2 ± 1.2 (*n* = 96) | 4.3 ± 3.1 (*n* = 100) | 8.8e−11 |
| TyG_index | 4.7 ± 0.2 (*n* = 96) | 5 ± 0.3 (*n* = 98) | 9.3e−13 |
| TG (mmol/L) | 1.4 ± 0.7 (*n* = 100) | 1.9 ± 1.2 (*n* = 98) | 9.2e−5 |
| Total Chol (mmol/L) | 5.6 ± 1.0 (*n* = 100) | 5.4 ± 1.0 (*n* = 98) | ns |
| LDL (mmol/L) | 3.8 ± 0.9 (*n* = 100) | 3.6 ± 0.9 (*n* = 98) | ns |
| HDL (mmol/L) | 1.6 ± 0.4 (*n* = 100) | 1.3 ± 0.4 (*n* = 98) | 6.5e−5 |
| CRP (mg/L) | 2.2 ± 2.7 (*n* = 100) | 3.5 ± 2.7 (*n* = 98) | 0.001 |
| WBC (×10⁹/L) | 5.8 ± 1.4 (*n* = 99) | 6.4 ± 2.4 (*n* = 98) | 0.04 |
| Neut (×10⁹/L) | 3.4 ± 1.0 (*n* = 99) | 3.8 ± 1.3 (*n* = 97) | 0.02 |
| Mono (×10⁹/L) | 0.4 ± 0.1 (*n* = 99) | 0.4 ± 0.1 (*n* = 97) | ns |
| EOS (×10⁹/L) | 0.2 ± 0.1 (*n* = 99) | 0.2 ± 0.1 (*n* = 97) | ns |
| Lymph (×10⁹/L) | 1.9 ± 0.5 (*n* = 99) | 1.9 ± 0.5 (*n* = 97) | ns |
| Statin | 4 (% av all) | 12 (% of all) | ns |
| PPI | 53 (% of all) | 54 (% of all) | ns |

Data are presented as mean ± SD, the *n* for each parameter is indicated within the parenthesis, Gender is presented as the percentage of females in the group, statin and PPI are presented as the percentage of users in the group. *P*-values indicate differences between the NGT and T2D group analyzed with two-tailed Mann–Whitney using Benjamini and Hochberg adjusted *p*-values. *Chol* cholesterol, *CRP* C-reactive protein, *DBP* diastolic blood pressure, *Eos* eosinophils, *FBG* fasting blood glucose, *HbA1c* Hemoglobin A1c, *HDL* high-density lipoprotein, *HOMA-IR* Homeostatic Model Assessment for Insulin Resistance, *LDL* low-density lipoprotein, *Lymph* lymphocytes, *Mono* monocytes, *NGT* normal glucose tolerance, *Neut* neutrophils, *OGTT 2 h BG* blood glucose at 2 h after oral glucose tolerance test, *PPI* proton-pump inhibitor, *SBP* systolic blood pressure, *T2D* Type 2 diabetes, *TG* triglycerides, *TyG* triglyceride-glucose index, *WBC* white blood cells, *whr* waist-to-hip ratio.

Droplet Digital PCR System (Bio-Rad) and amplification was performed by incubating at 95 °C for 5 min followed by 40 cycles of 94 °C for 30 s, 60 °C for 1 min, and then holding at 98 °C for 10 min to deactivate the enzyme. A ramp rate of 2 °C per second was used. Reactions were read using the QX200 Droplet reader (Bio-Rad) and analyzed using Quan-taSoft software v1.6.

## Bile acid metabolism in coculture

Overnight monocultures of bacterial community members were harvested by anaerobic centrifugation and resuspended in fresh pre-reduced Mega Medium with 100 µM CA (Sigma-Aldrich, St. Louis, MO, USA) and allowed to incubate at 37 °C for 48 h to induce bile acid metabolism. These cultures were harvested by anaerobic centrifugation, pellets were washed with pre-reduced PBS and resuspended densely (half the volume of the prime culture) in fresh pre-reduced

Mega Medium. Bacteroidetes and Firmicutes cultures, excluding *C. scindens*, were combined 1:2 by volume and aliquoted. *C. scindens* was serially diluted 1:10 into aliquots of the combined base community and CA was spiked into each coculture from a 100 mM DMSO stock solution to a final concentration of 100 µM. Cocultures were sampled for bile acid and DNA analyses 24 h post-CA spike. For bile acid analysis, 2 ml of culture was sampled and immediately acidified with 50 µL of 6 N HCl to stop all metabolic activity and protonate bile acids to make them more soluble in organic solvent. For DNA analysis, 1 ml of culture was sampled, pelleted, and stored at −80 °C until extraction and sequencing.

## Bile acid profiling of bacterial cultures by gas chromatography–mass spectrometry (GC-MS)

This method was applied to experiments presented in Fig. 1A–D. Acidified bacterial cultures were extracted twice with equal volumes of ethyl acetate, and the organic extracts were pooled, passed through a plug of Na₂SO₄, and concentrated to dryness. The residue was purified using a 1.5 cm silica pipet column. The column was washed with 2 ml of 5% methanol in dichloromethane, eluted with 6 ml of 9% methanol in ethyl acetate, and the eluent was concentrated to dryness. Prior to GC-MS analysis, purified bile acids were converted to their methyl ester–trimethylsilyl ether derivatives as follows. Bile acids were resuspended in 50 µL methanol and a 2 M solution of (trimethylsilyl)diazomethane in diethyl ether was added dropwise until a yellow color persisted. The solution was concentrated to dryness and 40 µL of a 3:1 solution of *N,O*-bis(-trimethylsilyl) trifluoroacetamide (BSTFA) and chlorotrimethylsilane was added. GC-MS analysis was performed on 3 µL of the resulting solution. Bile acids were identified based either on retention times and MS fragmentation patterns of standards or on published MS fragmentation patterns[34].

## Bile acid profiling of bacterial cultures by ultra-performance liquid chromatography (UPLC)–quadrupole time of flight mass spectrometry (Q-TOF-MS)

This method was adapted from a method developed by Justin R. Cross[35] and was applied to experiments presented in Fig. 1E. Briefly, acidified bacterial cultures were extracted twice with equal volumes of ethyl acetate, the organic extracts were pooled, passed through a plug of Na₂SO₄, concentrated to dryness and reconstituted in 50% methanol in water. Compounds were separated on an Agilent 1290 Infinity II UPLC using a Kinetex C18 column (1.7 µm, 2.1 × 100 mm, Phenomenex, Cat. #00D-4475-AN) and detected using an Agilent 6530 Q-TOF equipped with a dual-nebulizer ESI source in negative ionization mode, sheath gas temperature was 350 °C. Chromatographic separation was achieved at a flow rate of 0.35 ml/min with a 32-min gradient (*t* = 0 min, 25% B; *t* = 1 min, 25% B; *t* = 25 min, 75% B; *t* = 26 min, 100% B; *t* = 30 min, 100% B; *t* = 32 min, 25% B), where Mobile Phase A was water with 0.05% formic acid and Mobile Phase B was acetone with 0.05% formic acid.

## Bile acid profiling of samples from mice and humans by UPLC-MS/MS

This method was applied to experiments presented in Fig. 2C, D, Fig. 3A–G, Supplementary Fig. 1A, B and Supplementary Data. Bile acids in serum and caecum from mice, and plasma and faeces from the humans, were analyzed using UPLC-MS/MS and quantified using a combination of unlabeled standards and deuterium-labeled internal standards. 25 µL serum and 50–100 mg of caecum from the mice, and 50 µL plasma and 50–100 mg faeces from the human participants, were used for the analyses.

Serum and plasma samples were extracted with 10 volumes of methanol containing d4-TCA, d4-GCA, d4-GCDCA, d4-GUDCA, d4-GLCA, d4-UDCA, d4-CDCA, d4-LCA (50 nM of each). After 10 min of

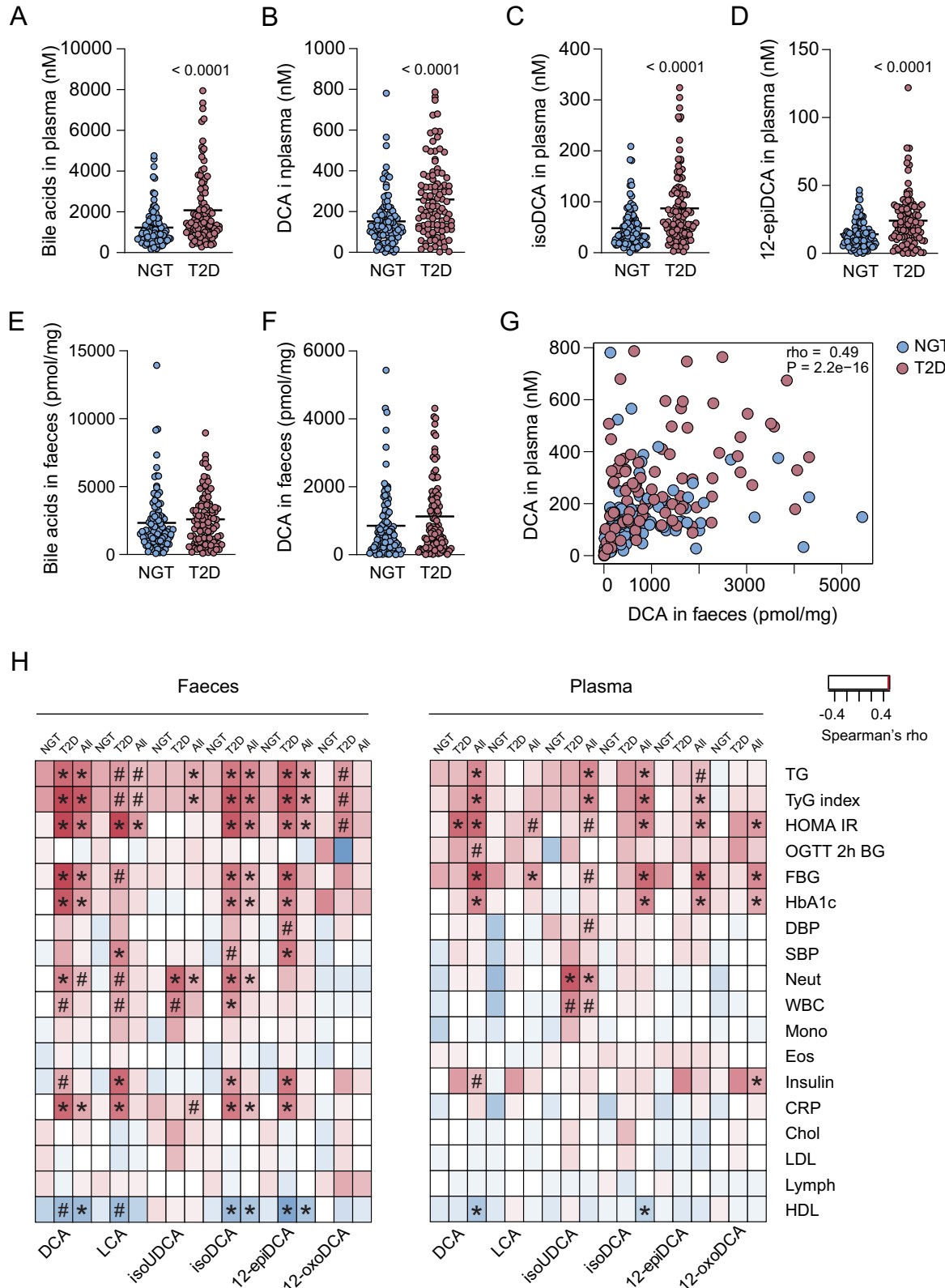

vortex and 10 min of centrifugation at 20,000 × g, supernatants were evaporated using a stream of nitrogen and reconstituted in 100 μL (for mouse serum) 200 μL (for human plasma) methanol:water [1:1]. 5 μL of the samples were used for the bile acid analysis.

Caecal and faecal samples were extracted with methanol, containing the same deuterium-labeled standards as above (2.5 μM of each). The samples were homogenized with ceramic beads for 10 min

using a Qiagen Tissuelyser II. After 10 min of centrifugation at 20,000 × g, 20 μL supernatant was diluted with 980 μL methanol:water 1:1.

UPLC-MS/MS conditions were according to previous work[36]. Briefly, after injection the bile acids were separated on a C18 column (1.7 μm, 2.1 ×100 mm; Kinetex, Phenomenex, USA) using water with 7.5 mM ammonium acetate and 0.019% formic acid (mobile phase A)

**Fig. 3 | Increased levels of DCA in plasma from T2D individuals. A–D** Plasma levels of total bile acids (**A**), DCA (**B**), isoDCA (**C**) and 12-epiDCA (**D**) in NGT and T2D individuals. **E, F** Faecal levels of total bile acids (**E**) and DCA (**F**) in NGT controls and T2D individuals. **G, H** Spearman's correlation analysis between DCA levels in plasma and faeces (**G**) and between DCA, LCA, isoUDCA, isoDCA, 12-epiDCA and 12-oxoDCA in faeces or plasma and clinical parameters (**H**). For (**A**–**F**), mean with all points are plotted; $n = 100$ samples/group; $P$-values indicate differences between the NGT (blue color) and T2D (pink color) groups analyzed with two-tailed Mann–Whitney using Benjamini and Hochberg adjusted $p$-values. For H, red color indicates positive correlation and blue color indicate negative correlation and

$^*P < 0.05$, $^\#P < 0.01$ are indicated using Benjamini and Hochberg adjusted $p$-values. Chol cholesterol, CRP C-reactive protein, DBP diastolic blood pressure, DCA deoxycholic acid, Eos eosinophils, FBG fasting blood glucose, HbA1c Hemoglobin A1c, HDL high-density lipoprotein, HOMA-IR Homeostatic Model Assessment for Insulin Resistance, isoDCA iso-deoxycholic acid, isoUDCA iso-ursodeoxycholic acid, LCA lithocholic acid, LDL low-density lipoprotein, Lymph lymphocytes, Mono monocytes, NGT normal glucose tolerance, Neut neutrophils, OGTT 2 h BG blood glucose at 2 h after oral glucose tolerance test, SBP systolic blood pressure, T2D Type 2 diabetes, TG triglycerides, TyG triglyceride-glucose index, WBC white blood cells, 12-epiDCA 12-epideoxycholic acid, 12-oxoDCA 12-oxodeoxycholic acid.

and acetonitrile with 0.1% formic acid (mobile phase B). The chromatographic separation started with 1-min isocratic separation at 20% B. The B-phase was then increased to 35% during 4 min. During the next 10 min the B-phase was increased to 100% and it was held at 100% for 3.5 min before returning to 20%. The total runtime was 20 min. Bile acids were detected using multiple reaction monitoring in negative mode on a QTRAP 5500 mass spectrometer (Sciex, Concord, Canada) and quantification was made against appropriate deuterated internal standards with adjustments by using external individual standard curves.

### Liver function tests
Liver transaminases, alanine transaminase (ALT) and aspartate transaminase (AST) were measured in 100 μl of serum from the mice. Analyses were performed at Clinical Chemistry Department, Sahlgrenska University Hospital, using the system Alinity ci 1303, software version 3.4.0 (Abbott laboratories).

### Quantitative real-time PCR of liver samples
Approximately 30 mg of livers from colonized mice were homogenized using Tissuelyser II (Qiagen) and total RNA was isolated using RNeasy mini kit (Qiagen, 74106). High-Capacity cDNA Reverse Transcription Kit (Applied Biosystems, 4368813) was used to synthesize 20 μl cDNA templates from 500 ng purified RNA using random hexamer primers, and the products were diluted 7x before use in subsequent reactions. 1x iQ™ SYBR® Green Supermix (Bio-Rad, 1708886) was used for qRT-PCR at final reaction volumes of 10 μl. 900 nM gene-specific primers were used in each reaction and all gene expression data was normalized to the *ribosomal protein L32* (*L32*) gene expression (forward: 5′-CCTCTGGTGA AGCCCAAGATC-3′; reverse: 5′-TCTGGGTTTCCGCCAGTTT-3′). Analyzed genes and primers were *cholesterol 7α-hydroxylase* (*Cyp7a1*) (forward: 5′-AGCAACTAAACAACCTGCCAGTACTA-3′; reverse: 5′-GTCCGGATATTCA AGGATGCA-3′) and *sterol 12α-hydroxylase* (*Cyp8b1*) (forward: 5′-GGCTGGCTTCCTGAGCTTATT-3′; reverse: 5′-ACTTCCTGAACAGCTCAT CGG-3′).

### Liver triglyceride measurements
Approximately 50 mg of livers from colonized mice were homogenized in 2 ml polypropylene tubes loaded with 6 zirconium oxide beads (3 mm) and then extracted using the BUME method[37]. Analysis of liver triglycerides was performed on an aliquot of the total extract using straight phase high-performance liquid chromatography (HPLC) as described previously[38] and quantification was made against an external calibration curve.

### Short-chain fatty acid analysis of caecal samples using GC-MS
Caecum short-chain fatty acids were measured using GC-MS. 100–150 mg of caecal content from colonized mice were mixed with internal standards, added to glass vials and freeze-dried. All samples were acidified with HCl, and short-chain fatty acids were extracted with two rounds of diethyl ether extraction. The organic supernatant was collected, the derivatization agent N-tert-butyldimethylsilyl-N-methyltrifluoroacetamide (Sigma-Aldrich) was added, and samples were incubated at room temperature overnight and short-chain fatty

acids were quantified with a 7090A gas chromatograph coupled to a 5975C mass spectrometer (Agilent Technologies). Isotopically labeled standards of propionic acid and sodium salts of acetate and butyrate were obtained from Sigma-Aldrich (Stockholm, Sweden). Quantification was made using a one-point calibration against the corresponding isotopically labeled internal standard. The samples were injected into an Agilent Technologies 7890A gas chromatograph equipped with a DB-5MS Ultra Inert column (30 m, 0.25 mm, 0.25 μm, Agilent Technologies) operated in split mode (50:1). Temperature gradient started at 70 °C for 3 min, changing to 200 °C within 9.5 min followed by a 30 °C/min. increase up to 280 °C and holding for 3.5 min, giving a total runtime of 15.7 min per sample. Analytes were detected using a mass spectrometer (Agilent Technologies 5975C) in electron ionization (EI) mode. For each analyte the most intense ion or fragment was selected for quantification.

### Statistical analyses
Differences between mice colonized with base community and base community + *C. scindens* were analyzed with Mann–Whitney for analysis of physiological parameters, and 2-way ANOVA with Šidák multiple comparisons test for analysis of the insulin tolerance test. Kruskal Wallis with Dunn's multiple comparison test was used when more than two groups were compared. Differences between the NGT and T2D group were analyzed with Mann–Whitney using Benjamini and Hochberg adjusted $p$-values. Spearman's correlation analysis with Benjamini and Hochberg adjusted $p$-values was used for correlation analysis. GraphPad Prism 9.5.0 was used for analysis of the in vivo data and R v4.3.2 was used for the analyses of the human data.

### Reporting summary
Further information on research design is available in the Nature Portfolio Reporting Summary linked to this article.

## Data availability
All data supporting the findings of this study are available within the paper and its Supplementary Information. Raw data for bile acid analysis of the human samples are provided in a Source data file. Phenotype data for the human participants can be requested following standard protocol for data access to the corresponding author. Source data are provided with this paper.

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

## Acknowledgements

We thank Anna Hallén, Carina Arvidsson, and Louise Helldén for technical assistance and the Adam Arkin Lab for providing *Eubacterium rectale* ATCC 33656. This work was supported by grants from the Leducq Foundation (17CVD01), the Swedish Heart Lung Foundation (20210366), the Knut and Alice Wallenberg Foundation (2017.0026), the Swedish Research Council (2019-01599), AFA insurances (160337), the Novo Nordisk Foundation (NNF21OC0070298), and from the Swedish state under the agreement between the Swedish government and the county councils, the ALF-agreement (ALFGBG-718101). F.B. is a Torsten Söderberg Professor in Medicine and Wallenberg Scholar. S.L.H. and M.A.F. were also partially supported by National Institutes of Health grant P01 HL147823.

## Author contributions

Conceptualization: A.W., A.B., M.A.F., and F.B. Clinical investigation: G.B. Methodology: A.W., A.B., L.O., W.S., A.L., and M.H. Formal analysis: A.W., L.O., H.W., and W.S. Visualization: A.W., A.B., and L.O. Resources: M.A.F., S.L.H., and F.B. Writing—original draft: A.W. Writing—review & editing: A.W., L.O., K.M., W.S., S.L.H., H.-U.M., M.A.F., and F.B.

## Funding

## Competing interests

Stanford University and the Chan Zuckerberg Biohub have patents pending for microbiome technologies on which A.B. and M.A.F. are co-inventors. M.A.F. is a co-founder and director of Federation Bio and Kelonia, a co-founder of Revolution Medicines, and a member of the scientific advisory boards of NGM Bio and Zymergen. S.L.H. reports having received royalty payments for inventions or discoveries related to cardiovascular diagnostics or therapeutics from Cleveland Heart Lab, a fully owned subsidiary of Quest Diagnostics, and Procter & Gamble,

being a paid consultant for Zehna Therapeutics, and having received research funds from Procter & Gamble, Pfizer Inc., Roche Diagnostics and Zehna Therapeutics. F.B. is co-founder and shareholder of Roxbiosens Inc and Implexion Pharma AB, receives research funding from Biogaia AB, and a member of the scientific advisory board of Bactolife A/S. The remaining authors declare no competing interests.
