## [Peer Review File · Nature Communications]

REVIEWERS' COMMENTS

Reviewer #2 (Remarks to the Author):

The authors' effort to highlight the remarkable potential of low-abundance bacteria in producing bioactive compounds is indeed commendable. Despite this, the revisions have not assuaged my reservations concerning the interactions between *C. sciendens*, its microbial counterparts, and their collective role in bile acid transformation. The mechanism behind the pronounced conversion rate—whether it stems solely from the *bai* genes and metabolic capabilities of *C. sciendens* or if it arises from a synergistic co-metabolism with surrounding bacteria—remains ambiguous.

While the authors have endeavored to highlight the methodological strengths of their study over previous research, my primary concerns have not been sufficiently addressed with substantial new evidence. As a result, the research primarily presents observational insights without delineating clear, novel mechanistic conclusions. In its current form, the study does not fulfill the rigorous criteria for publication in *Nature Communications*.

Reviewer #3 (Remarks to the Author):

Wahlstrom et al present a revised manuscript in which the previous critiques have been addressed.

REVIEWERS' COMMENTS

Reviewer #2 (Remarks to the Author):

The authors' effort to highlight the remarkable potential of low-abundance bacteria in producing bioactive compounds is indeed commendable. Despite this, the revisions have not assuaged my reservations concerning the interactions between *C. sciendens*, its microbial counterparts, and their collective role in bile acid transformation. The mechanism behind the pronounced conversion rate—whether it stems solely from the *bai* genes and metabolic capabilities of *C. sciendens* or if it arises from a synergistic co-metabolism with surrounding bacteria—remains ambiguous. While the authors have endeavored to highlight the methodological strengths of their study over previous research, my primary concerns have not been sufficiently addressed with substantial new evidence. As a result, the research primarily presents observational insights without delineating clear, novel mechanistic conclusions. In its current form, the study does not fulfill the rigorous criteria for publication in Nature Communications.

Response: We agree that we cannot exclude synergistic co-metabolism from other bacteria in the community and have now included the following text in the discussion and a new reference.

“We cannot exclude synergistic effects between *C. sciendens* and other bacteria in the gut, comparable to what we previously have shown between *Faecalibacterium prausnitzii* and *Desulfovibrio piger* (24). In that study, co-colonization of mice with *D. piger* increased the abundance and metabolic capacity of *F. prausnitzii* and it will be important to elucidate if similar synergistic co-metabolism with other bacteria in the community enhance DCA production from *C. sciendens*.”

Reviewer #3 (Remarks to the Author):

Wahlstrom et al present a revised manuscript in which the previous critiques have been addressed.

We thank the reviewer for his/her positive response.